# Electronic components embedded in a single graphene nanoribbon

P.H. Jacobse[1], A. Kimouche[2], T. Gebraad[3], M.M. Ervasti[2], J.M. Thijssen[3], P. Liljeroth [2] & I. Swart[1]

The use of graphene in electronic devices requires a band gap, which can be achieved by creating nanostructures such as graphene nanoribbons. A wide variety of atomically precise graphene nanoribbons can be prepared through on-surface synthesis, bringing the concept of graphene nanoribbon electronics closer to reality. For future applications it is beneficial to integrate contacts and more functionality directly into single ribbons by using hetero-structures. Here, we use the on-surface synthesis approach to fabricate a metal-semiconductor junction and a tunnel barrier in a single graphene nanoribbon consisting of 5- and 7-atom wide segments. We characterize the atomic scale geometry and electronic structure by combined atomic force microscopy, scanning tunneling microscopy, and conductance measurements complemented by density functional theory and transport calculations. These junctions are relevant for developing contacts in all-graphene nanoribbon devices and creating diodes and transistors, and act as a first step toward complete electronic devices built into a single graphene nanoribbon.

[1] Debye Institute for Nanomaterials Science, Utrecht University, P.O. Box 80000, Utrecht, 3508 TA The Netherlands. [2] Department of Applied Physics, Aalto University School of Science, P.O. Box 15100, 00076 Aalto, Finland. [3] Kavli Institute of Nanoscience Delft, Delft University of Technology, Lorentzweg 1, Delft, 2628 CJ The Netherlands. Correspondence and requests for materials should be addressed to I.S. (email: i.swart@uu.nl)

Rapid progress has been made in high-performance graphene devices for the study of new physical phenomena[1–5]. Mainstream applications, however, require a nonzero band gap[6, 7]. This effect can be introduced in a variety of ways, one of the most prominent being the preparation of narrow strips of graphene known as graphene nanoribbons (GNRs)[8–10]. The properties of these GNRs sensitively depends on width, orientation, and edge geometry[5, 9, 11–13]. Armchair-terminated GNRs are either metallic or semiconducting, depending on their exact atomic width: $3N$ or $3N+1$ (where $N$ is an integer) atom wide armchair GNRs are semiconductors, and those in the $3N+2$ family are metallic[5, 9, 14–18]. This extreme sensitivity to the detailed atomic structure also implies that traditional fabrication methods, such as e-beam lithography, are not precise enough to fabricate structures of sufficient quality. This limitation has been overcome with the on-surface, bottom-up synthesis of atomically precise GNRs. In this route, precursor molecules containing halogen atoms are evaporated onto a metal substrate, typically Au (111), and heated to form polymeric chains via Ullmann coupling. These chains are converted into fully aromatic GNRs through a cyclodehydrogenation step occurring at a higher temperature than the initial polymerization step[9].

The wide variety of atomically precise GNRs that can be prepared through on-surface synthesis[10] has brought the concept of GNR electronics closer to reality with the first prototype transistors having been demonstrated[19]. For future applications, direct integration of electrical contacts and functional electronic components such as diodes and tunnel barriers into a single ribbon would be highly advantageous. This can be realized by synthesizing GNR heterostructures using a combination of different precursor molecules. For example, semiconductor-semiconductor heterostructures in single GNRs have been demonstrated through synthesis from precursors that give rise to segments of different widths[20] or segments with substitutional nitrogen doping[21].

Here, we use the bottom-up approach to fabricate a metal-semiconductor junction and a tunnel barrier in a single GNR by utilizing atomically perfect connections between 5- and 7-atom wide segments (denoted as 5-GNR and 7-GNR, respectively). Not only are such junctions relevant for developing contacts in all-GNR devices, they also provide an additional route to create diodes and transistors[10, 19–21]. We characterize the atomic scale geometry and electronic structure by combined atomic force microscopy (AFM), scanning tunneling microscopy (STM), and conductance measurements. The GNR equivalent of a tunnel barrier constitutes a first step toward complete electronic devices built into a single GNR.

## Results

**Synthesis of GNR heterojunctions.** Figure 1a shows the precursors used in this study: 10,10′-dibromo-9,9′-bianthryl (DBBA) and 3,9-dibromoperylene (DBP). These precursors can be used to grow semiconducting (band gap of 2.7 eV) and metallic GNRs, respectively[9, 17]. Co-deposition of both precursors on Au(111) was used to prepare heterojunctions, as well as heterostructures (ribbons with more than one junction). The overview STM scan shown in Fig. 1b displays nanoribbons with a clear width modulation, corresponding to 5-GNR and 7-GNR segments, indicating successful copolymerization. A higher resolution AFM image of a longer ribbon with several 5-GNR and 7-GNR segments is shown in Fig. 1c. A heterojunction consisting of $n$ monomers of the 5-GNR precursor connected to $m$ monomers of the 7-GNR precursor is referred to as 5/7-GNR($n,m$).

Two distinct types of junctions (labeled I and II) were found, images of which are shown in Fig. 1d, e, respectively. The type I junction consists of a single six-membered ring between the collinear 5-GNR and 7-GNR segments. The enhanced contrast on one side of the anthracene moiety of the junction in the AFM image (Fig. 1d), as well as the increased apparent height at that location in the STM image, show that this junction is non-planar[22]. This non-planarity is caused by the steric repulsion between the inner hydrogen of the cove-edged helicene motif[23–26]. Junction II (Fig. 1e) consists of a hexagon and a pentagon (*red arrow* in Fig. 1a, *bottom*). The formation of the additional C–C bond results in a planarization of the junction and the introduction of an angle of 15° between the 5- and 7-GNR segments. Junction I can be converted to junction II by an additional cyclodehydrogenation reaction[27]. Post-annealing to $T = 600$ K results in a sample showing predominantly type II junctions. In the following, only ribbons with junctions of type II are considered.

**Electronic structure of the GNR heterojunctions.** We now turn to the electronic structure of the heterojunctions. The results of density functional theory (DFT) calculations on three types of gas-phase GNRs are shown in Fig. 2. To the *left* of the *vertical dotted line* in Fig. 2a, the orbital energies of a pure 7-GNR are given as a function of the number of repeating bisanthene units. The states at approx. −3.4 and −3.8 eV correspond to states localized on the two zigzag termini of the ribbon. Because of this localization, the energy of these states is virtually independent of the length of the $N = 7$ segment. The states on the zigzag termini exhibit a significant spin-splitting, i.e., one spin-channel is significantly lower in energy than the opposite channel[28]. In a pure $N = 7$ ribbon the

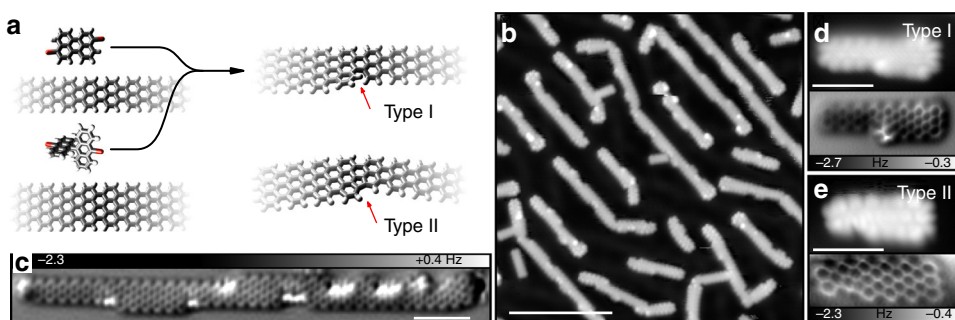

**Fig. 1** Fabrication and characterization of metal-semiconductor heterojunctions and heterostructures. **a** Chemical structure of 10,10′-dibromo-9,9′-bianthryl (DBBA) and dibromoperylene (DBP), and of a 5/7-GNR heterojunction. The *red arrows* indicate a type I (*top*) and type II (*bottom*) junction. **b** Overview STM image of 5/7-GNR heterostructures on Au(111). $V = 0.1$ V, $I = 20$ pA, scale bar is 10 nm. **c** AFM image of a heterostructure. **d** Small-scale STM (*top*) and AFM (*bottom*) images of junction type I. $z$-offset for AFM image—40 pm w.r.t. a STM set-point of $V = 0.1$ V, $I = 20$ pA. **e** Small-scale STM (*top*) and AFM (*bottom*) images of junction type II. $z$-offset—40 pm w.r.t. a STM set-point of $V = 0.1$ V, $I = 100$ pA. Scale bars in **c–e** are 2 nm

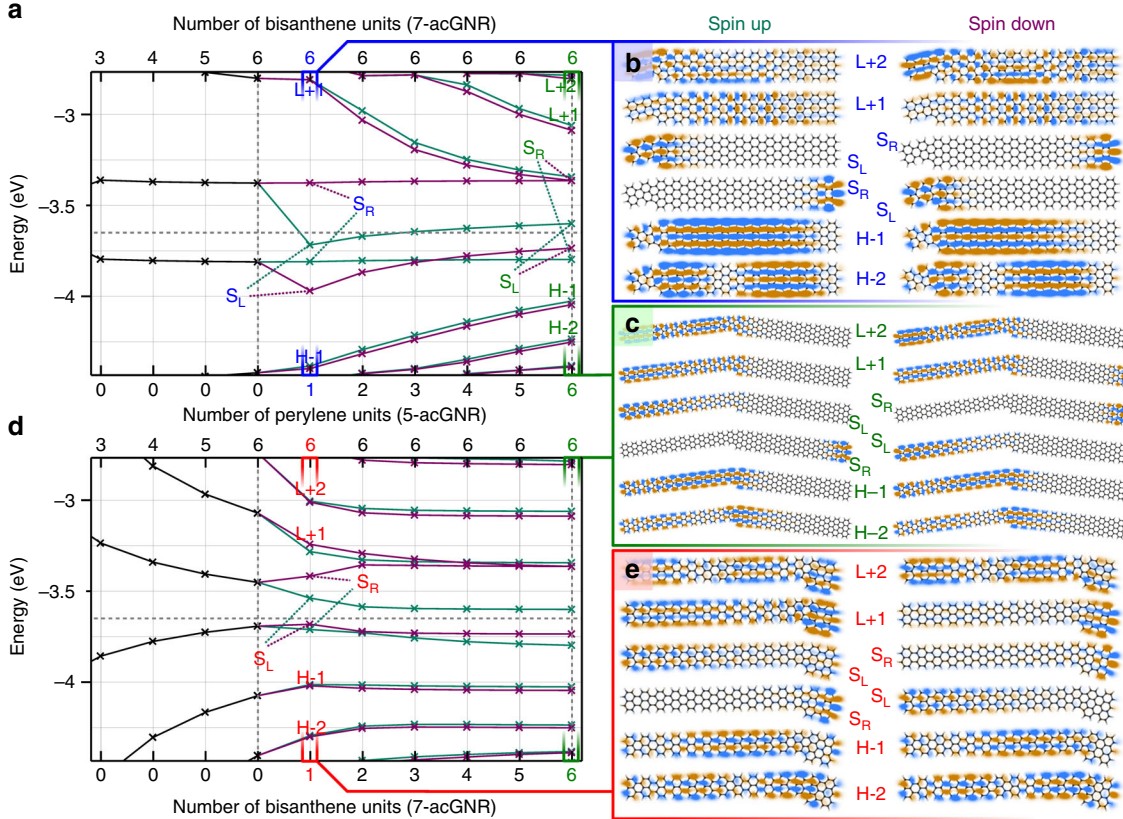

**Fig. 2** Electronic structure of 5/7-GNR heterojunctions calculated by DFT. **a** Evolution of the energy levels of pure 7-GNR as a function of length (*left* of the *vertical dotted line*), and of 5/7-GNR as a function of the number of 5-GNR monomers connected to a 7-GNR 6-mer. **b, c** Orbital plots of the six frontier molecular orbitals of the **b** 5/7-GNR(1,6) and **c** 5/7-GNR(6,6). "H" denotes the highest occupied molecular orbital (HOMO), "L" denotes the lowest unoccupied molecular orbital (LUMO), and "$S_{(L/R)}$" denotes a singly occupied molecular orbital localized on the left/right side (SOMO). **c** Same as **b** but for 5/7-GNR(6,6). Abbreviations as in **b**. **d** Evolution of the energy levels of pure 5-GNR (*left* of the *vertical dotted line*) as a function of length, and of 5/7-GNR as a function of the number of 7-GNR monomers connected to a 5-GNR 6-mer. **e** Orbital plots of the six frontier molecular orbitals of the 5/7-GNR(6,1). Abbreviations as in **b**

states localized on opposite termini are degenerate. However, upon connecting a single monomer of the 5-GNR, the degeneracy of the spin-polarized states is lifted (the two spin channels are represented by *purple* and *green lines*). As can be seen in Fig. 2b, the frontier orbitals of the perylene unit hybridize with the spin-split states localized on the zigzag end to which it is connected. Due to this hybridization, the spatial extent of these states increases, resulting in a lowering of the energy. In contrast, the spin states localized on the opposite zigzag end are unaffected. The spin splitting does not sensitively depend on the length of the 5-GNR part. Note that upon increasing the number of 5-GNR monomers, the energy gap between the bulk states decreases. As evidenced by the molecular orbitals of the 5/7-GNR(6,6) heterojunction, (Fig. 2c), these states are mostly localized on the narrow part of the heterojunction. States localized on the semiconductor segment of the ribbon are only found at higher and lower energy. Hence, the heterojunction can be characterized as a metal-semiconductor junction. The same picture emerges when connecting 7-GNR monomers to a pure 5-GNR nanoribbon (Fig. 2d, e). Upon attaching a bisanthene unit, the spin-degeneracy of the lowest unoccupied molecular orbital of the 5-GNR ribbon is lifted. The spin-splitting of these states rapidly converges to a constant value. We note that the splitting of the 7-GNR end states can also be explained in the framework of breaking of the bipartite symmetry of the lattice, see Supplementary Discussion.

Now, we turn our attention to the characterization of the junctions with scanning tunneling spectroscopy (STS) and differential conductance mapping. DFT calculations have been performed for all experimentally analyzed GNRs. The calculations do not take substrate effects into account. This should not affect the energetic order of the frontier orbitals. Figure 3a shows an STM topograph of a single junction 5/7-GNR(2,2) and a homogeneous ribbon segment 5-GNR(3). At low bias (0.1 V), there is enhanced contrast on the heterojunction compared to the pure 5-GNR (note the orbital structure and increased apparent height of the heterojunction). This indicates that the state originally localized at the zigzag edge of the 7-GNR hybridizes with states of the 5-GNR, in agreement with the DFT results. The differential conductance spectrum taken at the free zigzag end of the 7-atom wide part of the junction (Fig. 3a, spectrum 6, position indicated in the *inset*) shows a peak close to zero bias (~30 mV), similar to pure 7-GNRs[16]. A state at this energy is also observed on the 5-GNR segment (spectra 3 and 4). DFT calculations show that this peak originates from near-degenerate orbitals on the 7-GNR end and 5-GNR segment, the latter showing hybridization with the 7-GNR end state on the interface. A spectrum acquired at the interface only exhibits a feature at approximately −0.8 V. Experimental and simulated constant-height differential conductance maps corresponding to these voltages are shown in Fig. 3b. The molecular orbitals are not reproduced as such in the dI/dV maps due to the finite size of the tip and the presence of background non-resonant tunneling. Nevertheless, the individual states can still be clearly recognized in the density variations over the ribbons, and are in good agreement with the results from theory.

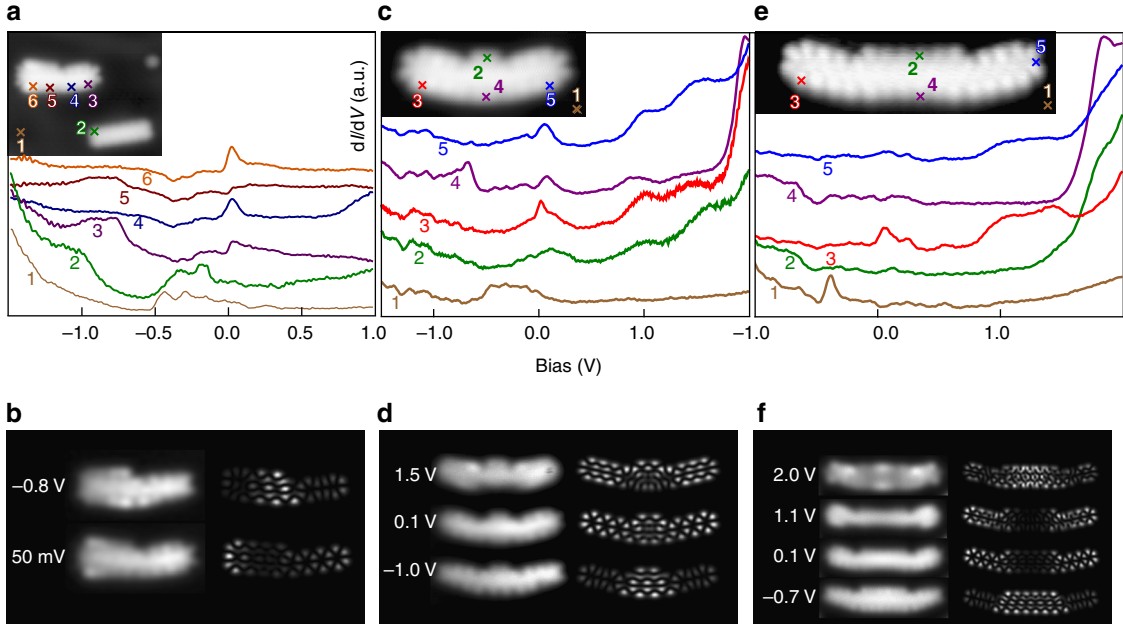

**Fig. 3** STS experiments and differential conductance maps of 5/7-GNR heterojunctions. **a** d$I$/d$V$ spectra acquired on a 5-GNR and 7-GNR segment of a 5/7-GNR(2,2) heterojunction. For comparison, a spectrum on a three monomer long pure 5-GNR is shown in *green*. Curves are vertically offset for clarity. *Inset*: STM topograph of the 5/7-GNR(2,2) heterojunction and neighboring pure 5-GNR ($V$ = 0.1 V, $I$ = 200 pA). **b** Experimental (*left*) and simulated (*right*) constant height d$I$/d$V$ maps recorded at $V$ = −0.8 V (*top*) and 50 mV (*bottom*), respectively. **c** d$I$/d$V$ spectra acquired on a 5/7/5-GNR(2,1,2) heterostructure. Curves are vertically offset for clarity. *Inset*: STM topograph of the 5/7/5-GNR(2,1,2) heterojunction ($V$ = 0.1 V, $I$ = 500 pA). **d** Experimental (*left*) and simulated (*right*) constant-height d$I$/d$V$ maps at $V$ = 1.5 V (*top*), $V$ = 0.1 V (*middle*), and $V$ = −1.0 V (*bottom*), respectively. **e** d$I$/d$V$ spectra acquired on a 5/7/5-GNR(2,3,2) heterostructure. Curves are vertically offset for clarity. *Inset*: STM topograph of the 5/7/5-GNR(2,3,2) heterojunction ($V$ = 0.1 V, $I$ = 100 pA). **f** Experimental (*left*) and simulated (*right*) constant-height d$I$/d$V$ maps at $V$ = 2.0 V, 1.1 V, 0.1 V, and −0.7 V, respectively

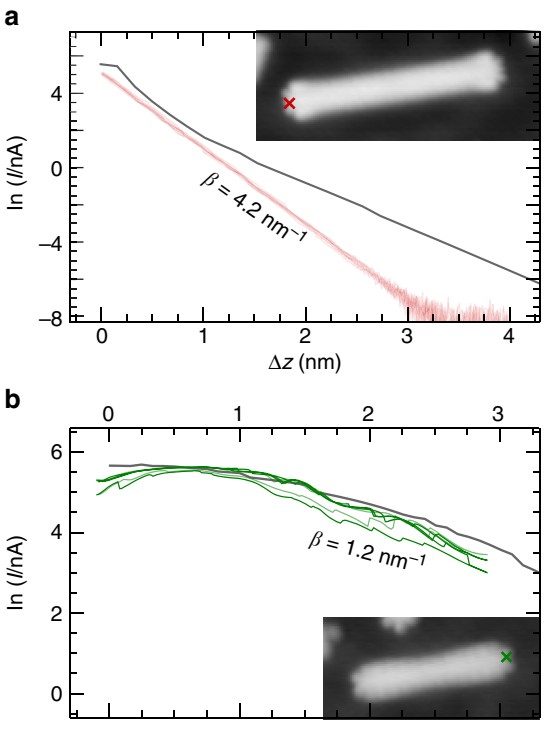

**Fig. 4** Conductance measurements on mono-component ribbons. **a** Experimental (*red*) and theoretical (*gray*) ln($I$/nA)($z$) curves for the 7-GNR(10) shown in the *inset*. The position from which the GNR was lifted is indicated by a *red cross*. **b** Experimental (*green*) and theoretical (*gray*) ln($I$/nA)($z$) curves for 5-GNR(6) shown in the *inset*. The position from which the GNR was lifted is indicated by a *green cross*

Experiments on longer heterostructures (Supplementary Note 1), as well as DFT calculations on the experimentally characterized junctions (Supplementary Note 2), support the same general picture: the low-bias states are mainly localized on the 5-GNR segment, whereas the 7-GNR part of the ribbon retains its semiconducting character. Additionally, the 7-GNR end state at the interface hybridizes with the 5-GNR bulk states.

In addition to single junctions, we studied ribbons with two junctions. Figures 3c, e show STM topographs of a 5/7/5-GNR (2,1,2) (short double junction) and a 5/7/5-GNR(2,3,2) (long double junction) heterostructures, respectively, and a set of d$I$/d$V$ spectra acquired at various positions above the ribbons. These structures represent a GNR equivalent of a tunnel barrier, where the semiconducting 7-GNR "channel" is sandwiched between metallic 5-GNR segments. Similar to the 5/7-GNR(2,2) junction, there is pronounced orbital character visible in the STM images taken at $V$ = 0.1 V. Indeed, all differential conductance spectra show a feature near 70 mV, except those recorded on the middle of the 7-GNR segment of the long double junction. This is also seen in the d$I$/d$V$ maps: in the short double junction (Fig. 3d, 0.1 V) the corresponding state is delocalized over the entire structure, while in the long double junction (Fig. 3f, 0.1 and 1.1 V), it is localized on the 5-GNR leads. For both ribbons, the first resonance at negative bias is localized on the 7-GNR segment (Fig. 3d, −1.0 V and Fig. 3f, −0.7 V). At positive bias, the first two states for the short ribbon (approx. 1 V and 1.5 V) are delocalized over the entire ribbon, whereas for the long ribbon these states are predominantly localized at the 5-GNR leads. We conclude that the low-bias, lead-localized states in the short double junction have too much overlap for the structure to be considered a tunnel barrier. The longer double junction has states localized on the leads with vanishing density in the 7-GNR segment, and this ribbon should, therefore, act as a tunnel barrier for low energy charge carriers.

**Charge transport through GNR heterojunctions**. We now discuss the conductance of these heterostructures. Conductance measurements on single GNRs were performed by picking up a nanoribbon with the STM tip and recording the current while retracting the tip. The measurements were corroborated by transport calculations (see Methods for details). Figure 4a shows experimental and simulated lifting curves for a pure 7-GNR(10). For 7-GNRs, pure exponential decay is observed: $I = I_0 e^{-\beta \Delta z}$, with $\beta = 4.2\ nm^{-1}$, in agreement with the result by Koch et al.[29]. Figure 4b shows a conductance measurement on pure 5-GNR(6). In all cases, there is an initial increase in the current before the current drops off again with $\beta \approx 1\ nm^{-1}$. The decay constant $\beta$ is proportional to the square root of the energy difference between the Fermi level (set by the bias voltage) and the frontier orbitals. As the gap for the 5-GNR segment is much smaller than that of the 7-GNR segment, the frontier orbitals of that segment will be closer to the Fermi level, resulting in a much smaller value of $\beta$[29]. The cause of the initial increase in the current with increasing tip-sample distance is unknown. One possible explanation is that the electronic coupling with the substrate decreases as the ribbon is lifted from the surface. This decrease in the coupling strength could result in a shift of the energy of an orbital, i.e., a transport channel, toward the bias voltage at which the lifting experiments were performed. For both types of GNRs, the results from the transport calculations are in good agreement with the experimental data. Slight differences between the experimental and simulated curves can be due to the limitations of the underlying tight binding model, as discussed in the Supplementary Discussion.

Figure 5a shows the current decay as a 5/7-GNR(3,4) heterojunction is lifted from the 5-GNR and 7-GNR termini (green and red, positions indicated in the inset images by crosses), respectively, and similarly for a 5/7-GNR(3,3) heterojunction lifted from the 7-GNR terminus (orange). When the ribbon is lifted starting with the 5-GNR segment, the current initially increases with increasing tip-sample distance, as observed for pure 5-GNRs. At larger distances the current decays exponentially, suggesting non-resonant (in-gap) electron transport. In this specific nanoribbon, the 5-GNR segment is so short that the fast current decay through the 7-GNR already appears before the characteristic initial increase is completed. Therefore, we cannot identify a monoexponential regime of the 5-GNR segment from which we can extract a value of $\beta$. When the ribbon is lifted from the 7-GNR side, the current first decays with $\beta \approx 4\ nm^{-1}$, followed by a regime with $\beta \approx 1\ nm^{-1}$. These values are close to those of the pure 7- and 5-GNRs, respectively. This result suggests that, as soon as the entire 7-GNR segment is lifted from the surface, the further reduction in the current with distance is dominated by the detachment of the 5-GNR segment. The experimentally observed behavior is qualitatively reproduced by the transport calculations (Fig. 5a, *bottom panel*). This current decay behavior is found in all single junctions (Supplementary Note 3).

Conductance measurements on 5/7/5-GNR(2,1,3) (short double junction) and 5/7/5-GNR(2,4,2) (long double junction) are shown in Fig. 5b, c. All double junctions show a signature of fast decay during the 7-GNR segment lift-off. For the short double junction, the length of the two 5-GNR segments differs by one 5-GNR monomer. As a result, the fast decay due to the lift-off of the 7-GNR segment occurs early when the ribbon is lifted from the short 5-GNR lead (1 nm < $\Delta z$ < 1.5 nm) and late when the ribbon is lifted from the long 5-GNR lead ($\Delta z$ > 2.3 nm). The long double junction (5/7/5-GNR(2,4,2)) has a very clear signature of slow decay during the 5-GNR lift-off ($\Delta z$ < 1 nm and $\Delta z$ > 3 nm) and fast decay during the 7-GNR lift-off (1 nm < $\Delta z$ < 3 nm). Again, the experimentally observed features are qualitatively reproduced in the transport calculations. The lift-off experiments were complemented by recording $I(V)$ curves for a number of

increasing tip heights (Fig. 5d–f), up to the point where the nanoribbon completely detached from the surface (evidenced by a sudden drop in the current and failure to relocate the nanoribbon on the surface in subsequent STM scans). The $I(V)$ curves for the single junction (Fig. 5d) show an onset of resonant tunneling at both sides of the bias window, as well as a relatively small peak in the current at low bias (around 500 mV). Since this peak occurs at energies that are in-gap with respect to the 7-GNR segment, it can be ascribed to tunneling from the tip—through the 7-GNR segment—into orbitals localized on the 5-GNR segment. The $I(V)$ curves of long and short double junctions (Fig. 5e, f) show a striking difference. The short junction has no gap, demonstrating that the states from the 5-GNR segments couple through the short 7-GNR segment. In contrast, the long double junction exhibits a wide bias range (−0.6 V to 0.1 V) where the current is nearly zero, i.e., it operates as a tunnel barrier. The spectra show a clear low-bias resonance at ∼0.2–0.4 V and two resonances at ∼0.8–1.4 V, which result from the low-bias lead-localized states as described above. The final two spectra ($\Delta z$ = 5.8 and 5.9 nm) exhibit negative differential conductance. At these heights, the coupling of the heterostructure to the surface is small and comparable to the coupling to the STM tip, restoring the degeneracy of the lead-localized states. At low bias, electrons can tunnel resonantly between the 5-GNR leads, resulting in an enhanced tunnel current. However, at higher bias, the degeneracy of the orbitals on 5-GNR segments will be lifted, resulting in a suppression of resonant tunneling and, therefore, in negative differential conductance. This result provides conclusive evidence that the 5/7/5-GNR(2,4,2) functions as a tunnel barrier.

## Discussion

The motivation for synthesizing GNRs through on-surface covalent coupling of molecular precursors is to develop atomically precise components that could be the key enabling component in next-generation nanoelectronics. We build on this idea by encoding a complete electrical device (e.g., tunnel barrier or a diode) in a single GNR with a feature size of ca. 1 nm with atomic precision and directly test its function through a transport measurement.

We fabricated GNR heterostructures consisting of segments of two different widths with totally different electronic properties: 7-atom wide semiconducting and 5-atom wide metallic GNR-segments. This allowed us to realize metal-semiconductor and metal-semiconductor-metal junctions embedded in a single GNR. We characterized the geometry of these heterostructures with atomically resolved AFM and probed their local density of states with STM and STS. We find that the electronic structure close to the Fermi level arises from hybridization of the 5-GNR states with the localized zigzag end states of the 7-GNR segments. This picture remains valid in more complicated heterostructures consisting of multiple junctions.

The electrical performance of the GNRs can only be properly understood based on transport measurements. We have taken steps towards this by carrying out STM-based two-terminal transport experiments on GNR heterostructures with a known atomic structure. These experiments demonstrate that the segments of different widths have strongly differing tunneling decay constants that are directly linked to their energy band gaps. Characterization of GNR heterostructures demonstrated that already four monomer units of 7-GNR can act as a tunnel barrier and effectively decouple 5-GNR leads from each other. STS measurements corroborated with DFT calculations further showed that this heterojunction with a semi-conducting channel composed of four bisanthene units has low transmission in the energy window between −0.8 V and 1.8 V, where the electron density is exclusively localized on the leads. As a result, this junction represents a true tunnel barrier between leads.

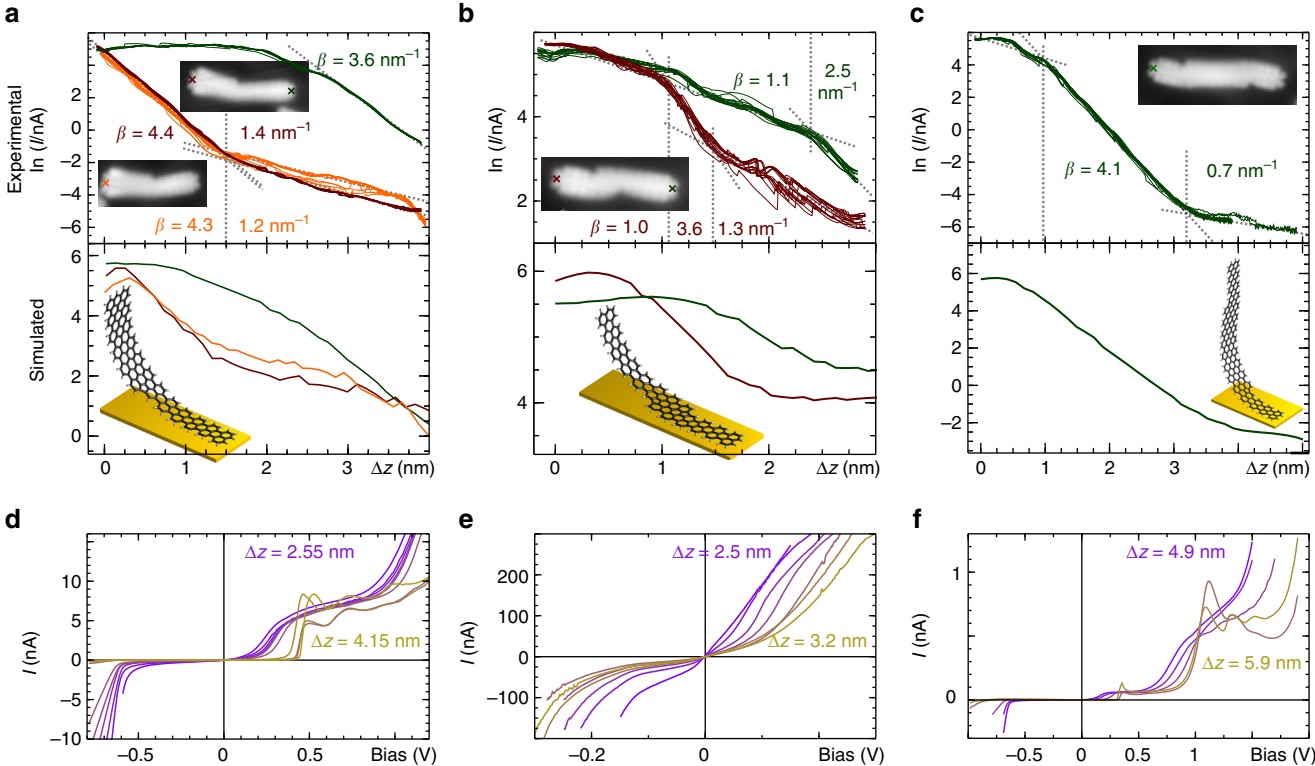

**Fig. 5** Conductance measurements of single GNR heterojunctions and heterostructures. **a** Experimental (*top*) and simulated (*bottom*) $\ln(I/nA)(\Delta z)$ curves obtained while lifting the 5/7-GNR(3,3) from the 7-GNR side (*orange*) and while lifting the 5/7-GNR(4,3) from the 5-GNR side (*green*) and 7-GNR side (*red*). The *insets* in the *top* figure show STM topographs of the respective ribbons ($V = 0.1$ V, $I = 40$ pA) with tip position for lifting experiment indicated by crosses, whereas the *inset* in the *bottom* image shows a calculated lifting geometry of the 5/7-GNR(3,3). **b** Same as **a**, but for lifting a 5/7/5-GNR(2,1,3) from the long 5-GNR side (*green*) and the short 5-GNR side (*red*). **c** Same as **a**, but for lifting a 5/7/5-GNR(2,4,2). **d–f** Experimental $I(V)$ curves recorded with increasing tip heights $\Delta z$ for 5/7-GNR(3,3) and 5/7/5-GNR(2,1,3) and 5/7/5-GNR(2,4,2) heterojunctions, respectively

Before real-life applications can be realized, our methods need to be further refined. Most importantly, the lengths of the different segments have to be controlled precisely and deterministically. This can conceivably be achieved through consecutive precursor supply, as a 1D analogue of the epitaxy in two-dimensional space[30–32]. An alternative method would be to use a hierarchical synthesis using different halogen atoms, where the Ullmann coupling reactions of the different monomers take place at different temperatures[33]. In addition, in order to contact the active parts of the GNR heterostructures with macroscopic leads, structures should be complemented by metallic parts. This could be achieved either through growth of longer metallic segments, or by incorporating 2D graphene islands[34]. Finally, methods for large-scale transfer of the GNR heterostructures from the growth substrate into device structures with positional control must be developed.

In conclusion, we demonstrated a complete electrical device contained in a single GNR, which represents a significant step toward all graphene nanostructure-based electronics. Moreover, the integration of metal-semiconductor heterojunctions in a single nanoribbon provides a seamless platform for realizing a broad range of multifunctional nanoscale electronic devices. The tunability of the on-surface synthesis through utilizing different molecular building blocks allows near-unlimited incorporation of atomically sharp interfaces and doping profiles into the GNR heterostructures.

## Methods
**STM and AFM measurements**. Samples were prepared by simultaneously evaporating the precursor molecules from two effusion cells onto a clean Au(111) single crystal held at $T = 480$ K. After maintaining the sample at this temperature

for 10–30 min, it was annealed at $T = 570$ K for 5 min to induce cyclodehydrogenation reactions. The sample was then inserted into a Scienta-Omicron low-temperature ($T = 4.5$ K) STM/AFM, housed within the same ultrahigh vacuum system (base pressure $< 5 \times 10^{-10}$ mbar). The qPlus sensor had a resonance frequency $f_0$ of 19.5 kHz, a quality factor of 30,000 and a spring constant $k = 1800$ N/m, and an oscillation amplitude of approximately 100 pm. CO terminated tips were prepared by picking up a CO molecule from the Au(111) surface using a method developed for Cu(111)[35, 36]. AFM images were acquired at $V = 0$ V. $dI/dV$ signals were recorded by a lock-in amplifier (frequency = 273 Hz, amplitude = 50 mV (r. m.s.) and 100 mV (r.m.s.) for the $dI/dV$ maps on the heterostructures).

**Conductance measurements**. The nanoribbon to be lifted was scanned in STM feedback mode, after which the tip was positioned above clean Au(111) next to the ribbon. The feedback setpoint was set to 400 pA at $V = 100$ mV. After switching the feedback off, the current was decreased to 10 pA and the tip was repositioned on the terminus of the ribbon. While monitoring the current in real time, the tip was slowly lowered toward the surface in 1 pm decrements. Successful attachment is evident from a sudden jump in the current from tens of pA to 200–300 pA, and typically occurs when the tip is lowered to $\Delta z = (-100 \pm 50)$ pm with respect to the feedback set point. From this position, the ribbon was lifted and $I(\Delta z)$ was recorded, or the tip was retracted to the desired height for $I(V)$ experiments. For most $I(\Delta z)$ experiments, the ribbon was lifted and lowered a number of times, until the nanoribbon detached or it was deemed that enough data were acquired.

**DFT calculations**. All DFT calculations were performed on free nanoribbons using ORCA, Program Version 3.0.2[37]. The Perdew–Burke–Ernzerhof exchange-correlation functional was used, together with a 6–31 g* basis set. Geometry optimizations were considered to be converged when the change in energy was smaller than $5 \times 10^{-6}$ a.u. and the forces on all atoms were below $3 \times 10^{-4}$ a.u.

**Transport calculations**. The electronic structure of the nanoribbon was modeled by a tight-binding model with on-site energy $\varepsilon_0 = -5$ eV, nearest-neighbor hopping of $t = 2.8$ eV and nearest-neighbor overlap parameter $s = 0.15$[38, 39]. The coupling to the metallic surface and tip was modeled by the self-energy method, where the C–Au coupling constants were weighed according to the C–Au distances. The geometry of the nanoribbons was computed by minimizing the mechanical energy

as a function of curvature for ribbons attached to the tip. Finally, the current was calculated by the Landauer formula[40]. Further details on the calculations are given in the Supplementary Methods.

**Data availability**. The data that support the plots within this paper and other findings of this study are available from the corresponding author upon reasonable request.

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

## Acknowledgements

We thank R. Bulo for giving us access to her computer cluster and D. Vanmaekelbergh for useful comments on the manuscript. This research was supported by the European Research Council (ERC-2011-StG No. 278698 "PRECISE-NANO"), the Academy of Finland (Centre of Excellence in Low Temperature Quantum Phenomena and Devices No. 284594), and the Debye Graduate School.

## Author contributions

P.L. and I.S. conceived the experiments, which were performed by P.H.J. and A.K. The experimental data were evaluated by P.H.J., A.K., P.L. and I.S. DFT calculations were performed by P.H.J. and I.S. and M.M.E. contributed to the DFT calculations on the heterostructures. The tight binding and transport calculations were performed by T.G., J.M.T., and P.H.J. All authors contributed to writing the manuscript.
