## [Peer Review File · Nature Communications]

Reviewers' comments:

Reviewer #1 (Remarks to the Author):

The manuscript has investigated electrical structure of different types of graphene nanoribbons (GNR) including 5-GNR, 7-GNR, and 5/7-GNR heterojunctions using both AFM, STM technique experimentally and DFT calculation theoretically. The authors also claim that a tunneling barrier is formed in 5/7-GNR heterojunctions based on conductance measurements through STM tip. The theoretical calculation and experimental data are clear and the main findings are easy to understand for the community. However, I found several confusing problems in reviewing this manuscript. Overall, I think the quality of the paper needs to be improved to fulfill the criteria to be considered for publication in Nature Communication based on the following suggestions/comments:

1. I'm wondering what is the main topic of this paper: analyzing the electrical structure of GNR or the demonstration of the GNR heterojunction, since half of the main content is on analyzing the electrical structure of GNR rather than GNR electronic components.

2. There are at least three published papers focusing on similar topics:

Cai, J. et al. Graphene nanoribbon heterojunctions. *Nat. Nanotechnol.* 9, 896-900 (2014) Chen, Y. et al. Molecular bandgap engineering of bottom-up synthesized graphene nanoribbon heterojunctions. *Nat. Nanotechnol.* 10, 156-160 (2015)

Kimouche, A. et al. Ultra-narrow metallic armchair graphene nanoribbons. *Nat. Commun.* 6, 10177 (2015)

The main findings and methods used in this manuscript are similar to, especially the third *Nature Commun.* paper. The novelty is not very obvious. It should be considered as incremental work.

3. The description and illustration need to be clearer. For example, in Figure 2, it would be better to explain what do L+2, SL, H-1... represent. Some data shown in the figure is not discussed or even mentioned in the manuscript itself, such as Figure 5(d). Also, there is no obvious relationship I could find in presenting Figure 4 if the main topic is focusing on GNR heterojunction. This part is more appropriate to be discussed in the SI part.

Overall, I cannot recommend publication of this manuscript in the present form.

Reviewer #2 (Remarks to the Author):

This manuscript reports on a very original piece of work, showing successful fabrication of metal-semiconductor junction and a tunnel barrier in a single GNR consisting of 5- and 7-atoms segments. Using scanning probe microscopy techniques authors characterize their structural and electronic properties. Transport measurements are applied here to test GNRs usefulness as complete electronic device. Experimental results are supported by DFT calculations. The research strategy is strong, the results are very clearly presented and their interpretation is convincing. Given the wide interest on GNRs and on their future implementation into devices, this article is expected to attract the attention of a large community of researchers. Only some minor points require further clarification/justification :

- Fig.3 page 7: In consideration of color-blind people (although also as an improvement for non color-blind people) I would suggest to mark consecutive dI/dV curves with numbers for clarity on each data set next to the curves and to the points in the insets.

- Page 8, paragraph on dI/dV maps: The text about the spatial distribution of the conductance maps in

the double heterojunctions is in nice agreement with the calculations. However, while the selected spectra support that description, it is not visible from the experimental conductance maps. Conductance profiles across the double heterojunction should be added to the figure to make the claimed effects visible.

- Figure 4a: Experimental and predicted β fit almost perfectly until $\Delta z \approx 1$. From then on, the predicted slope changes and differs from the experimental one. Do the authors have a hypothesis for that mismatch?

- Fig.4b page 9: experimental data (for 5-GNR) show initial increase in the current. Authors should propose an interpretation for this phenomenology.

- Figure 5a: when the single junction is lifted from the 5-GNR segment (green line) one doesn't see a slope $\beta \approx 1.2$ like in figure 4b and 5b. This should be mentioned in the text and, if possible, propose an explanation for it.

- Figure 5b and 5c: a description of the current decay variations for the double junctions would be appreciated. Only the central region of the long heterojunction is addressed in the text, but the rest is not described. Also an explanation on the asymmetry in 5b depending on which side is lifted would be greatly appreciated.

- Methods: Authors claim that CO terminated tips were prepared as described previously and refer to Cu substrate. Did the Authors pick up CO molecule directly from Au surface? If yes, they should describe the methodology.

In addition, please correct the following typos.

- Paragraph "in addition (...).(page 8) Second sentence: "Figure 3c and 3d (...). It's not correct should be Figure 3c and 3e.

- In the same paragraph (page 8). Sentence: "This is also seen in the dI/dV maps: in the short double junction (Fig.3e middle row)(...). Should be (Fig.3d middle row). Authors should check the text carefully and fix all the editorial bugs

- Page 8: "For both ribbons, the first resonance at negative bias is localized on the 7-GNR segment (bottom rows in Fig.3e (should be 3d) and 3f)". For the short ribbon, experimental and simulated data are consistent but for the long ribbon it's not so obvious.

In conclusion Authors present state-of-art results proving that obtained GNR heterostructures can be considered as a potential building-blocks of novel nanoelectronic devices. I thus suggest acceptance of this manuscript in Nature Communication once these points have been addressed.

Reviewer #3 (Remarks to the Author):

This manuscript reports a detailed experimental investigation of the assembly and characterization of GNR junctions. The work is adequately complemented by a dedicated theoretical analysis based on density functional calculations. In addition, the reported study includes the measurement of electronic transport properties using a "lift-off" technique based on a specific use of STM.

The field of "bottom-up" GNR synthesis is an extremely active one, following the pioneering paper

from Cai et al in 2010. Since then, a number of groups have worked to reproduce the Cai's results and have moved beyond single-GNRs, including junctions. This work is the latest addition to the existing literature. However, this work is not a mere incremental development on existing work. The originality of the work consists in:

1. Well defined junctions between metallic and semiconducting GNRs (most of the prior works have been limited to junctions between semiconducting GNRs).
2. The study of "sandwiched" systems, akin to tunnel junctions.
3. The experimental determination of transport properties of the junctions.

The paper is particularly well and logically written and I have not identified any specific and critical weakness. In other words, it is recommended that this paper be published. However, the authors should consider the following points:

1. To my knowledge this work is the first to identify the presence of pentagonal rings at the interface between GNRs (in contrast, pentagons present on the edges have been reported repeatedly). I would think that the presence of the pentagon at the junction should have important impact since its effect is to break the electron-hole symmetry, and therefore significantly affect the transport properties. Maybe the authors should discuss this some more?
2. The authors do not talk much at all about substrate effects. Yet, it has been broadly demonstrated that GNR's electronic states are affected by the substrate. Of course, their technical approach to measure transport does not suffer from this since the GNRs are lift-off the substrate. However, the STM and STS are affected by this, especially since they are compared to "gas-phase" calculations. The authors should make a note about that.
3. It is unclear how much control there is to build the junctions. It seems to be largely a stochastic method where structures are created "by chance". The authors should discuss this apparent challenge and also potentially provide means to develop mitigating strategies.

Electronic components embedded in a single graphene nanoribbon

Point-by-point response

Reviewer #1

We thank the referee for reviewing our manuscript, affirming its clarity and comprehensibility, and for her/his suggestions on how to improve the manuscript.

Comment: I'm wondering what is the main topic of this paper: analyzing the electrical structure of GNR or the demonstration of the GNR heterojunction, since half of the main content is on analyzing the electrical structure of GNR rather than GNR electronic components.

Response: The main topic of the paper is the synthesis and characterization of metal-semiconductor junctions embedded in graphene nanoribbons. This involves a discussion on the synthesis through copolymerization of the precursor molecules, as well as an electronic characterization of the as-formed GNR through scanning tunneling spectroscopy and conductance experiments, and corroborated by theoretical calculations. We think our manuscript presents a complete picture of the fabrication and characterization of GNR heterostructures.

Action: none

Comment: There are at least three published papers focusing on similar topics: Cai, J. et al. Graphene nanoribbon heterojunctions. Nat. Nanotechnol. 9, 896-900 (2014) , Chen, Y. et al. Molecular bandgap engineering of bottom-up synthesized graphene nanoribbon heterojunctions. Nat. Nanotechnol. 10, 156–160 (2015), Kimouche, A. et al. Ultra-narrow metallic armchair graphene nanoribbons. Nat. Commun. 6, 10177 (2015). The main findings and methods used in this manuscript are similar to, especially the third Nature Commun. paper. The novelty is not very obvious. It should be considered as incremental work.

Response: The first two papers mentioned by the referee report on the synthesis of semiconductor-semiconductor nanoribbon heterojunctions and have been cited in our manuscript. The third paper describes the synthesis of pure 5-acGNR. That article partly forms the basis for our work: we use the GNRs described by Kimouche et al and combine them with the well-known 7-acGNRs to generate *metal*-semiconductor junctions. These junctions are fundamentally different from semiconductor-semiconductor junctions. Furthermore, we report a detailed investigation on the atomic structure of the junctions and perform, for the first time, lifting experiments on nanoribbon heterojunctions. In particular the lifting experiments on graphene nanoribbons incorporating heterojunctions are unprecedented in literature. Hence, the new elements of our work are (i) preparation of *metal*-semiconductor heterojunctions embedded in graphene nanoribbons (ii) characterization of their geometric and electronic structure (iii) the fact that we have performed two-terminal transport experiments on graphene nanoribbon heterojunctions. These points are also recognized by the other two referees.

Action: none

Comment: The description and illustration need to be clearer. For example, in Figure 2, it would be better to explain what do L+2, SL, H-1... represent.

Response: The referee is correct in noting that the abbreviations in Figure 2 have not been properly introduced. The abbreviations “H”, “S”, and “L” stand for HOMO, LUMO and SOMO, respectively, whereas the subscript L/R denotes whether the SOMO is localized on the left or right side of the GNR.

Action: In order to clarify the abbreviations in Figure 2, we have added the following sentence to the caption in the description of panel b:

“H” denotes the highest occupied molecular orbital (HOMO), “L” denotes the lowest unoccupied molecular orbital (LUMO), and “S_(L/R)” denotes a singly occupied molecular orbital localized on the left/right side (SOMO).

Also, for panels c and e the captions were extended to read ‘Abbreviations as in b.’

Comment: Some data shown in the figure is not discussed or even mentioned in the manuscript itself, such as Figure 5(d).

Response: We agree with the referee that Figure 5D was not properly discussed in the original version of the manuscript.

Action: We have added the following sentence in order to give a proper description of the I(V) measurements presented in Figure 5d:

The I(V) curves for the single junction (Figure 5d) show an onset of resonant tunneling at both sides of the bias window, as well as a relatively small peak in the current at low bias (around 500 mV). Since this peak occurs at energies that are in-gap with respect to the 7-GNR segment, it can be ascribed to tunneling from the tip – through the 7-GNR segment – into orbitals localized on the 5-GNR segment.

Comment: Also, there is no obvious relationship I could find in presenting Figure 4 if the main topic is focusing on GNR heterojunction. This part is more appropriate to be discussed in the SI part.

Response: Thus far I(z) conductance experiments have only been reported for 7-acGNR. To fully appreciate the results reported for the heterostructures, the reader needs to understand the results obtained for the 5-acGNR. We therefore believe it is beneficial for the broad readership of Nature Communications to include the results presented in Figure 4 in the main text, also considering the fact that the manuscript does not exceed any length restrictions (as far as we are aware).

Action: none

Reviewer #2

We thank the referee for his careful reading and positive remarks on our work, and for his/her suggestions on how to improve the manuscript.

Comment: Fig.3 page 7: In consideration of color-blind people (although also as an improvement for non color-blind people) I would suggest to mark consecutive dI/dV curves with numbers for clarity on each data set next to the curves and to the points in the insets.

Response: We thank the referee for this very useful suggestion.

Action: We have modified Figure 3 as suggested by the referee.

Comment: Page 8, paragraph on dI/dV maps: The text about the spatial distribution of the conductance maps in the double heterojunctions is in nice agreement with the calculations. However, while the selected spectra support that description, it is not visible from the experimental conductance maps. Conductance profiles across the double heterojunction should be added to the figure to make the claimed effects visible.

Response: We agree with the referee insofar that we realize that the density variations in the dI/dV maps are quite subtle, and the orbital pattern can be hard to recognize. Several factors can contribute, i.e. the finite size of the tip, as well as “non-resonant” (background) tunneling. Finally, in the experiment, the contours of the local density of states are probed at a (much) lower contour value than the simulated maps shown in the original version of the manuscript (i.e. larger tip-sample distance). Nevertheless, the intensity variations are in fact visible from the maps. In particular, the $-1.0V$ map in Figure 3d shows a significantly increased orbital density in the bottom of the 7-GNR segment as compared to the map at $0.1V$, whereas the $1.5V$ map shows a clear decrease in density in the same area. The maps at $0.1V$ and $1.1V$ in Figure 3f show an increased density on the 5-GNR leads, whereas the map at $-0.7V$ has its largest density on the 7-GNR segment.

We find that adding conductance profiles to the maps do not improve the readability of the figure. In contrast, we feel that adding more curves makes the figure more confusing, without significantly improving the clarity of the spatial electron density variations. We think the most clear way of interpreting the differential conductance maps is still to look at the density variations, and to compare them with maps recorded at different bias voltages and simulated maps. Even though the variations are not very pronounced at first sight, a closer look should still be enough to convince the reader that the maps are in agreement with spectroscopy experiments and theory.

Action: In order to convey the message that the orbital density only shows up in a subtle fashion in the differential conductance maps due to aforementioned effects, we have added the following text:

The molecular orbitals are not reproduced as such in the dI/dV maps due to the finite size of the tip and the presence of background non-resonant tunneling. Nevertheless, the individual states can still be clearly

recognized in the density variations over the ribbons, and are in good agreement with the results from theory.

Comment: Figure 4a: Experimental and predicted β fit almost perfectly until $\Delta z \approx 1$. From then on, the predicted slope changes and differs from the experimental one. Do the authors have a hypothesis for that mismatch?

Response: Indeed, there is a mismatch between theory and experiment. In a more general sense, we see that the qualitative features of the $I(z)$ curves are nicely reproduced by the calculations, but the quantitative values of the current decay are often underestimated. Any discrepancies are most likely due to the limitations of the tight binding model in describing the complicated electronic structure of the junction during the lifting experiments (bending of the ribbon, modifications of the contact with the substrate). In order to better convey this point, we have added a brief discussion to the SI.

Action: We have added the following paragraph to the SI, explaining what we believe to be the limiting factors in the transport calculations:

We note that in the calculations, the current decay parameter is often underestimated. We believe this to be due to the absence of explicit electron-electron interactions in the tight binding model, which results in an underestimation of the band gap. In the case of non-resonant transport, this error translates into an underestimation of β , since the energetic distance between the Fermi level of the tip and the HOMO and LUMO is reduced. An additional effect is that the end-localized state may contribute to the transport, but due to its confined nature, its contribution may quickly decay upon lifting. We believe this to be the reason why in the calculation of the conductance in Figure 4a, β is higher at the beginning of the $I(z)$ curve, where the end state still plays a role. After lifting to $\Delta z > 1$ nm, the current decay converges to a monoexponential decay, the β value of which underestimates the real value due to the underestimated band gap. The fact that the same end state feature cannot be seen in the experiments could be due to the ribbon already being slightly lifted from the substrate upon being contacted, meaning that the zero height is shifted with respect to the zero in the calculations. Unfortunately, we cannot measure the absolute value of the tip height in STM.

We have referred to this explanation in the manuscript after introducing the transport through pure 5-GNR and 7-GNR. Here, the text reads as follows:

Slight differences between the experimental and simulated curves can be due to the limitations of the underlying tight binding model, as discussed in Supplementary Section 5.

Comment: Fig.4b page 9: experimental data (for 5-GNR) show initial increase in the current. Authors should propose an interpretation for this phenomenology.

Response: This effect is indeed surprising and counterintuitive.

Action: The following text was added to include our hypothesis of the initial current increase effect:

The cause of the initial increase in the current with increasing tip-sample distance is unknown. One possible explanation is that the electronic coupling with the substrate decreases as the ribbon is lifted from the surface. This decrease in the coupling strength could result in a shift of the energy of an orbital, i.e. a transport channel, towards the bias voltage at which the lifting experiments were performed.

Comment: Figure 5a: when the single junction is lifted from the 5-GNR segment (green line) one doesn't see a slope $\beta \approx 1.2 \text{ nm}$ like in figure 4b and 5b. This should be mentioned in the text and, if possible, propose an explanation for it.

Response: In the same way as seen in the 5-GNR, the 5/7-GNR lifted on the 5-GNR side displays an initial increase in the $I(z)$ curves before the exponential decay kicks in. This is the same effect as the one discussed in the previous comment. This effect prevents us from extracting a beta parameter as a monoexponential decay is not obtained for the 5-GNR segment before the 7-GNR segment is lifted. This is noteworthy, and mentioned in the revised version.

Action: The following text was added:

In this specific nanoribbon, the 5-GNR segment is so short that the fast current decay through the 7-GNR already appears before the characteristic initial increase is completed. Therefore, we cannot identify a monoexponential regime of the 5-GNR segment from which we can extract a value of β .

Comment: Figure 5b and 5c: a description of the current decay variations for the double junctions would be appreciated. Only the central region of the long heterojunction is addressed in the text, but the rest is not described. Also an explanation on the asymmetry in 5b depending on which side is lifted would be greatly appreciated.

Response: We agree with the referee that the current decay variations should be discussed more. We have added text in order to clarify the effects in the $I(z)$ curves and improve the readability.

Action: The following text was added:

All double junctions show a signature of fast decay during the 7-GNR segment lift-off. For the short double junction, the length of the two 5-GNR segments differs by one 5-GNR monomer. As a result, the fast decay due to the lift-off of the 7-GNR segment occurs for smaller values of Δz when the ribbon is lifted from the short 5-GNR lead ($1 \text{ nm} < \Delta z < 1.5 \text{ nm}$) and at larger values when lifted from the long 5-GNR lead ($\Delta z > 2.3 \text{ nm}$). The long double junction (5/7/5-GNR(2,4,2)) has a very clear signature of slow decay during the 5-GNR lift-off ($\Delta z < 1 \text{ nm}$ and $\Delta z > 3 \text{ nm}$) and fast decay during the 7-GNR lift-off ($1 \text{ nm} < \Delta z < 3 \text{ nm}$).

Comment: Methods: Authors claim that CO terminated tips were prepared as described previously and refer to Cu substrate. Did the Authors pick up CO molecule directly from Au surface? If yes, they should describe the methodology.

Response: Yes, the CO molecule was picked up directly from the Au(111) surface. The method that was developed for Cu(111) also works very well for Au(111).

Action: In the methods section, the relevant sentence now reads:

CO terminated tips were prepared by picking up a CO molecule from the Au(111) surface using a method developed for Cu(111).^{34,35}

Comment: In addition, please correct the following typos.

- Paragraph "in addition (...).(page 8) Second sentence: "Figure 3c and 3d (...). It's not correct should be Figure 3c and 3e.

- In the same paragraph (page 8). Sentence: "This is also seen in the dI/dV maps: in the short double junction (Fig.3e middle row)(...). Should be (Fig.3d middle row). Authors should check the text carefully and fix all the editorial bugs

- Page 8: "For both ribbons, the first resonance at negative bias is localized on the 7-GNR segment (bottom rows in Fig.3e (should be 3d) and 3f)". For the short ribbon, experimental and simulated data are consistent but for the long ribbon it's not so obvious.

Response: We thank the referee for pointing out these errors in the manuscript.

With respect to the third remark: Spectra 2 and 4, which were both acquired on the 7-GNR segment in Figure 3e show a clear onset at negative bias (-0.7V) whereas spectra taken on the 5-acGNR part (3 and 5) remain flat.

Action: The errors have been corrected.

Reviewer #3

We thank the referee for his/her positive comments and feedback on the manuscript.

Comment: To my knowledge this work is the first to identify the presence of pentagonal rings at the interface between GNRs (in contrast, pentagons present on the edges have been reported repeatedly). I would think that the presence of the pentagon at the junction should have important impact since its effect is to break the electron-hole symmetry, and therefore significantly affect the transport properties. Maybe the authors should discuss this some more?

Response: We thank the referee for this interesting insight.

Action: We have incorporated a discussion on the effect of the five-membered ring, based on tight binding calculations in the supplementary information. This discussion is referenced in the manuscript by the addition of the following sentence:

We note that the splitting of the 7-GNR end states can also be explained in the framework of breaking of the bipartite symmetry of the lattice, as shown in Supplementary Fig. 7.

Comment: The authors do not talk much at all about substrate effects. Yet, it has been broadly demonstrated that GNR's electronic states are affected by the substrate. Of course, their technical approach to measure transport does not suffer from this since the GNRs are lift-off the substrate. However, the STM and STS are affected by this, especially since they are compared to "gas-phase" calculations. The authors should make a note about that.

Response: It has indeed be shown that the electronic coupling with the underlying Au(111) significantly affects the electronic structure of GNRs. It has been shown that screening/image charge effects may lower the observed band gap of GNR. We therefore not rely on the energy values of the DFT, but rather on the order of the orbitals, which should be more robust.

Action: The following text has been added:

Now, we turn our attention to the characterization of the junctions with scanning tunneling spectroscopy (STS) and differential conductance mapping. DFT calculations have been performed for all experimentally analyzed GNRs. The calculations do not take substrate effects into account. This should not affect the energetic order of the frontier orbitals.

Comment: It is unclear how much control there is to build the junctions. It seems to be largely a stochastic method where structures are created "by chance". The authors should discuss this apparent challenge and also potentially provide means to develop mitigating strategies.

Response: Indeed, the presented work deals with a stochastic process. As noted in the discussion section, in order to realize real-life applications, the growth of the nanoribbons should be further

controlled. We pointed out that consecutive precursor supply during growth could be investigated as a method to obtain nanoribbons with a more isotropic distribution of 5-GNR and 7-GNR segments. An additional method to increase control over heterojunction formation is to exploit a hierarchical synthesis strategy, for example by using different halogen atoms. This would allow the growth of segments of the first type, after which growth at elevated temperature activates the second precursor for Ullmann coupling. We agree with the referee that we could have been more elaborate on the process, and we have therefore appended this insight into the text.

Action: The following text was added:

An alternative method would be to use a hierarchical synthesis using different halogen atoms, where the Ullmann coupling reactions of the different monomers take place at different temperatures.

REVIEWERS' COMMENTS:

Reviewer #2 (Remarks to the Author):

The revised version has been satisfactorily improved and all my minor concerns addressed. I recommend publication of the article in Nature Communications.

Reviewer #3 (Remarks to the Author):

The authors have carefully considered all the points made in my initial report.

(Note that the bipartition is actually broken by the presence of the pentagon, which, in reality, connect the two sub lattices)

I can now fully support publication of this work in Nat. Comm.